# Cellulose Nanocrystals Show Anti-Adherent and Anti-Biofilm Properties against Oral Microorganisms

**DOI:** 10.3390/bioengineering11040355

**Published:** 2024-04-05

**Authors:** Antonella Panio, Andrei C. Ionescu, Barbara La Ferla, Luca Zoia, Paolo Savadori, Gianluca M. Tartaglia, Eugenio Brambilla

**Affiliations:** 1Oral Microbiology and Biomaterials Laboratory, Department of Biomedical, Surgical, and Dental Sciences, University of Milan, Via Pascal, 36, 20133 Milan, Italy; antonella.panio@guest.unimi.it (A.P.); eugenio.brambilla@unimi.it (E.B.); 2Fondazione IRCCS Cà Granda Ospedale Maggiore Policlinico, 20100 Milan, Italy; paolo.savadori@unimi.it (P.S.); gianluca.tartaglia@unimi.it (G.M.T.); 3Department of Earth and Environmental Sciences, Università degli Studi di Milano-Bicocca, Piazza dell’Ateneo Nuovo, 1, 20126 Milan, Italy; barbara.laferla@unimib.it (B.L.F.); luca.zoia@unimib.it (L.Z.); 4Department of Biomedical, Surgical, and Dental Sciences, University of Milan, Via Della Commenda, 10/12, 20122 Milan, Italy

**Keywords:** cellulose, nanoparticles, bacterial adherence, biofilms, *Streptococcus mutans*, *Candida albicans*, bacteria viability, bioreactors, scanning electron microscopy

## Abstract

Cellulose nanocrystals (CNCs) are cellulose-derived nanomaterials that can be easily obtained, e.g., from vegetable waste produced by circular economies. They show promising antimicrobial activity and an absence of side effects and toxicity. This study investigated the ability of CNCs to reduce microbial adherence and biofilm formation using in vitro microbiological models reproducing the oral environment. Microbial adherence by microbial strains of oral interest, *Streptococcus mutans* and *Candida albicans,* was evaluated on the surfaces of salivary pellicle-coated enamel disks in the presence of different aqueous solutions of CNCs. The anti-biofilm activity of the same CNC solutions was tested against *S. mutans* and an oral microcosm model based on mixed plaque inoculum using a continuous-flow bioreactor. Results showed the excellent anti-adherent activity of the CNCs against the tested strains from the lowest concentration tested (0.032 wt. %, *p* < 0.001). Such activity was significantly higher against *S. mutans* than against *C. albicans* (*p* < 0.01), suggesting a selective anti-adherent activity against pathogenic strains. At the same time, there was a minimal, albeit significant, anti-biofilm activity (0.5 and 4 wt. % CNC solution for *S. mutans* and oral microcosm, respectively, *p* = 0.01). This makes CNCs particularly interesting as anticaries agents, encouraging their use in the oral field.

## 1. Introduction

Bacterial adherence and biofilm formation inhibition are increasingly promising topics because of the urgent need for alternative strategies to conventional antimicrobial principles [1]. Within this context are situated the main strategies for counteracting dental caries and periodontal disease [2,3]. In the oral environment, active principles must interact positively with the surfaces of natural and artificial hard tissues, modulating the biofilm behavior instead of eradicating its structures. In this context, anti-adherence and biofilm inhibition strategies could effectively restore the plaque ecosystem’s balance [4,5,6]. From this point of view, one of the most promising strategies is based on using a wide variety of nanoparticles, taking advantage of their ability to interact with both the microorganisms and their colonization surface [7].

In this sense, cellulose nanocrystals (CNCs) represent a valid possibility. They are cellulose-derived nanomaterials that can be easily obtained [8] because cellulose is one of the most diffused vegetable wastes produced by circular economies [9]. CNCs are widely available and easily produced through acidic hydrolysis from many widely available cellulose sources, including agricultural and agro-industrial residues, such as wood, non-wood fibers, and algae [10,11,12].

CNCs show many desirable properties, such as excellent mechanical strength, absence of toxicity, and, above all, biocompatibility [11,13]. These characteristics make them worthwhile for several biomedical applications.

As stated above, the most interesting property of CNCs is their antibacterial potential [14,15], expressed against bacterial adherence and biofilm formation, without any notable side effects. These characteristics fit the general requirements of an active principle that can be used to prevent biofilm-related oral diseases. In particular, the absence of known side effects and toxicity [15,16,17,18] seems ideal for oral hygiene products such as toothpaste and mouth rinses. For this reason, CNCs have attracted the curiosity of those in the oral field looking for antimicrobial agents.

Such agents target a broad spectrum of oral microorganisms, with particular attention to cariogenic species, such as *Streptococcus mutans*, and, for different reasons, related to mucosae infections, *Candida albicans*. *S. mutans*, in particular, is a species that exhibits intense cariogenicity and biofilm formation capability, and is, for this reason, a model microorganism for in vitro caries testing [19,20]. Furthermore, growth conditions represent a crucial parameter when looking to provide in vitro data with good translatability to clinical conditions. For these reasons, when testing the antimicrobial properties of new active principles for the oral environment, it is of utmost importance that the two main phases of microbial colonization (adherence and biofilm formation) are considered and tested in an environment with hydrodynamic stress similar to the natural one [21].

This study aimed to evaluate the ability of a CNC solution to reduce bacterial adherence and biofilm formation using in vitro microbiological models reproducing different aspects of the oral environment.

## 2. Materials and Methods

### 2.1. Preparation and Characterization of CNCs

According to a previously reported procedure [22], CNCs were obtained by acid hydrolysis, which includes treating Whatman#1 filter paper (GE Healthcare, Buckinghamshire, UK) with 64% of H_2_SO_4_ at 55 °C for 1 h. The reaction was quenched by adding distilled water (10 times the acid volume). The product was centrifuged (3000× *g*, 4 °C, 15 min) and dialyzed (cut-off 14,000 g/mol) to remove residual acid. The CNCs were finally recovered by centrifugation (12,000× *g*, 4 °C, 30 min), and a 12 wt. % stock solution was prepared. Characterization was performed as follows. Samples were prepared by diluting the previously obtained stock solution to a 140 mg/L concentration of CNC in ultrapure water. Attenuated total reflectance-Fourier transform infrared spectroscopy (ATR-FTIR) spectra were collected on a few mg of oven-dry sample using an iTR Smart device connected to a Nicolet iS10 spectrometer (Thermo Fisher Scientific, Cleveland, OH, USA). The instrument was set up to perform 32 consecutive scans, with a resolution of 2 cm^−1^ and a range of 4000–800 cm^−1^. An operating laser with a wavelength of 632.8 nm and a backscattering angle of 173° was applied to conduct DLS and ζ-potential measurements on a Zeta Sizer Nano 3600 (Malvern Panalytical Ltd., Malvern, UK) instrument. Collected data were digested using the proprietary software (Zetasizer 7.03, Malvern Panalytical Ltd., Malvern, UK). Measurements were carried out at 25 °C using a disposable cuvette. Three replicate measurements per sample were performed to establish measurement repeatability. Then, scanning electron microscopy (field emission gun scanning electron microscope, FEG-SEM, Zeiss Ultra Plus, Jena, Germany, operated at 7.0 kV accelerating voltage) was used to ascertain the nanocrystals’ morphology and size. Samples were prepared by dispersing 10 μL of the stock solution in 3 mL of ultrapure water. The mixture was sonicated for 10 min, then 1 mL was removed, dispersed in 9 mL of absolute ethanol, and sonicated again for 10 min. A total of 100 μL of the CNC ethanol suspension was deposited on a stub for analysis and air-dried. Ultrastructural analysis was investigated by transmission electron microscopy (TEM) using a Zeiss LEO 912AB (Carl Zeiss Microscopy GmbH, Oberkochen, Germany) operating at 120 kV, where displayed images were digitalized using a CCD-BM/1K system. A total of 10 μL of the previously prepared CNC ethanol suspension was dropped onto Formvar-coated 300 mesh copper grids and air-dried. Samples were counterstained for 5 min with a saturated solution of uranyl acetate, washed with water, and allowed to air-dry.

### 2.2. Procedures

Serial dilutions of the stock solution were prepared as 4%, 2%, 1%, 0.5%, 0.25%, 0.125%, and 0.063% by dilution using filter-sterile water, while the control solution was filter-sterile water.

### 2.3. Preparation of Enamel Disks

One hundred and twenty-six enamel–dentin disks measuring 6.0 mm in diameter and 2.0 mm in thickness were produced from anterior adult bovine teeth. The labial surfaces were prepared by slicing them with a water-cooled trephine diamond bur (INDIAM, Carrara, Italy). We used a polishing machine (Motopol 8; Buehler, Düsseldorf, Germany) and 600/1000/4000-grit grinding paper (Buehler, Lake Bluff, IL, USA) to remove dentin bottoms and standardize the polishing process for the enamel surfaces. In preparation for the tests, a chemiclave that featured hydrogen peroxide plasma technology (Sterrad; ASP, Irvine, CA, USA) was used to sterilize all the disks. No heat-related damage to the specimens occurred since the highest temperature was limited to 45 °C [23].

#### 2.3.1. Saliva Preparation

Experimenters (ACI, AP, and PS) were asked to expectorate their stimulated whole saliva. They did not have any cavities, had not taken antibiotics for at least three months, and did not brush their teeth during the 24 h period leading up to the experiment. The methods were based on the approach described in the 2001 publication by Guggenheim et al. [24]. After collecting saliva in refrigerated tubes, it was mixed and then heated to 60 °C for 30 min to deactivate endogenous enzymes. After that, it was centrifuged at 12,000× *g*, 4 °C, for 15 min. After transferring the supernatant to sterile tubes, it was frozen at −20 °C and thawed at 37 °C for 1 h before the tests. After 24 h, the same experimenters collected their plaque from the labial area of the upper molars, mixed it with whole saliva, and then used it as inoculum. This was done to create the artificial oral microcosm [25].

#### 2.3.2. Microorganisms

A previously described protocol was followed for cultivating *Streptococcus mutans* ATCC 35668, as stated in [26]. In short, *S. mutans* was inoculated onto Mitis Salivarius Bacitracin agar and incubated at 37 °C in a 5% supplemented CO_2_ environment for 48 h. One single colony was then inoculated in brain heart infusion and additioned with 1 wt. % sucrose at 37 °C in a 5% enriched CO_2_ atmosphere for 12 h (early log phase) to generate a pure microorganism suspension. After being separated by centrifugation at a speed of 2200× *g* and a temperature of 19 °C for 5 min, the *S. mutans* cells were rinsed twice with sterile phosphate-buffered saline (PBS) and then mixed with BHI additioned with 1 wt. % sucrose. Then, the suspension was run through a Sonifier type B-150 (Branson, Danbury, CT, USA) with a 7 W energy output for 30 s to break up the bacterial chains. The microbial concentration was estimated to be about 6.0 × 10^8^ cells/mL after the suspension was adjusted to a 1.0 on the McFarland scale.

After 24 h of incubation at 37 °C in a 5% enriched CO_2_ atmosphere, a pure culture of the *Candida albicans* strain ATCC 90028 in BHI + 1 wt. % sucrose was obtained. After centrifuging at 2200× *g* at 19 °C for 5 min, the cells were rinsed twice with sterile PBS. They were then resuspended in BHI + 1 wt. % sucrose to achieve a turbidity level of 1.0 on the McFarland scale.

#### 2.3.3. Adherence

Enamel disks were placed with sterile tweezers, one into each well of 48-well plates (Nunc, Kastrup, Denmark), and 50 µL of sterile thawed saliva was added on the surface of each disk. An acquired salivary pellicle was allowed to develop on the enamel surfaces during incubation at 37 °C in a 5% supplemented CO_2_ atmosphere for 24 h. After that, excess saliva was discarded, and the surface of the disks was washed with sterile PBS. Then, 250 µL of each concentration of CNCs was added to each of the *n* = 8 wells. Subsequently, 250 µL of either *S. mutans* or *C. albicans* microbial suspension was added to each well. Thus, the final concentration of CNCs in each well was 2.00%, 1.00%, 0.50%, 0.25%, 0.12%, 0.06%, 0.03%, and 0% (negative control). A total of 64 disks for each strain were used. We rinsed the enamel disks using sterile PBS at 37 °C after 2 h of incubation to remove non-adhered microbial cells and then measured the biomass viability (Section 2.4).

#### 2.3.4. Biofilm Formation

The use of a modified-drip flow reactor (M-DFR) to assess biofilm development under shear stress conditions was documented by Ionescu et al. 2019 [27]. Thanks to the redesign, we were able to submerge the enamel disks entirely in the flowing media by using customized specimen trays to keep the disks on the bottom of the flow cells (Figure 1). Before the experiments began, we used the Sterrad chemiclave to disinfect all the tubing and trays holding the disks. After being put together in a sterile hood (Figure 2), the M-DFR was moved to an incubator and set to run at 37 °C. For 24 h, the enamel disk surfaces of every flow cell (*n* = 6) were exposed to the thawed sterile saliva to replicate the formation of a salivary pellicle. Afterward, any surplus saliva was thrown away. To achieve biofilm growth on the enamel disk surfaces, 10 mL of *S. mutans* or pooled whole saliva suspension was added to each flow cell. Each strain used 64 disks, equally divided among three flow cells. The flow cells were supplied with a modified artificial saliva medium at a consistent rate after 4 h by means of a multi-channel, computer-controlled peristaltic pump (RP-1; Rainin, Emeryville, CA, USA). The modified artificial saliva medium was prepared in a sterile environment. It included the following ingredients: 10.0 g/L sucrose, 2.5 g/L mucin (type II, porcine gastric), 2.0 g/L tryptone, 2.0 g/L bacteriological peptone, 1.0 g/L yeast extract, 0.35 g/L sodium chloride, 0.2 g/L calcium chloride, 0.2 g/L potassium chloride, 0.1 g/L cysteine hydrochloride, 1 mg/L hemin, and 0.2 mg/L vitamin K_1_. A flow rate of 20.0 mL/h was established. After 24 h, the medium flow was turned off, the flow cells were opened, and the specimen trays were delicately removed and promptly put in 37 °C Petri dishes filled with sterile PBS. The next step was to carefully transfer the enamel disks from the tray to 48-well plates. Each disk was exposed to 500 μL of previously diluted CNC solution (2%, 1.00%, 0.50%, 0.25%, 0.12%, 0.06%, 0.03%, and 0%, *n* = 8 disks for each dilution). Plates were incubated at 37 °C in a 5% supplemented CO_2_ environment for 2 h. Then, the solutions were discarded, and the wells were washed with sterile PBS at 37 °C to remove loosely attached cells and to avoid the influence of residual colloidal CNC solutions on the following absorbance readings. Finally, the biomass viability was assessed on the surface of the enamel disks as follows (Section 2.4).

### 2.4. Biomass Viability Assay

Following the procedures outlined above, the amount of adherent and viable (metabolically active) biomass was analyzed [27]. In brief, 5-(4,5)-dimethylthiazol-2-yl-2,5-diphenyltetrazolium bromide (MTT) and 0.3 mg/mL of N-methylfenazinium methyl sulfate (PMS) were dissolved in sterile PBS to make two stock solutions that were held at 2 °C in light-proof vials. On the day of the experiment, a fresh measurement solution (FMS) was made by diluting the MTT stock solution, PMS stock solution, and sterile PBS at a ratio of 1:1:8, respectively. Dissolving 10 vol % sodium dodecyl sulfate and 50 vol % dimethylformamide in ultrapure water produced a lysing solution (LS).

Each disk-containing well (both from adherence and biofilm formation experiments) received 300 µL of FMS solution, and the plates were left to incubate at 37 °C in a dark environment for 1 h. Microbes’ redox processes and, mainly, electron transport across their plasma membranes transformed the soluble yellow MTT salt into purple formazan crystals that precipitated during incubation. The PMS served as an intermediary electron acceptor, easing the conversion process at the cell membrane level. After delicately removing the unreacted FMS solution, 300 µL of LS was added to each well to dissolve all precipitated formazan crystals. The plates were kept at 37 °C in a dark environment for 1 h and then were gently shaken for 10 min, and 100 µL of solution from each well was transferred to 96-well plates. Spectrophotometer readings were taken at 550 nm using a Genesys 10-S instrument (Thermo Spectronic, Rochester, NY, USA) to determine the solution’s absorbance. The quantity of adhering, viable, and metabolically active biomass was represented by the relative absorbance in optical density (OD) units. By setting the negative control (filter-sterile water) at 100% and the positive control (a 1 wt. % chlorhexidine digluconate solution in filtered sterile water) at 0%, the results were transformed to % viability.

### 2.5. Scanning Electron Microscopy (SEM) Evaluation

An additional four specimens for each CNC dilution (inoculated with *S. mutans* for 2 h and 24 h) were obtained as previously described. After rinsing with sterile PBS, specimens were placed into a freshly prepared Karnovsky’s fixative solution for 2 h (2% paraformaldehyde and 2% glutaraldehyde in 0.1 M sodium cacodylate buffer). After an additional rinsing in cacodylate buffer, the specimens were passed through a graded ethanol series (35, 50, 70, 80, 90, and 100 vol%). To remove the ethanol and, at the same time, minimize shrinkage, specimens underwent critical point drying (Critical-Point Dryer, EMS 850, Hatfield, PA, USA). After that, they were mounted on stubs with conductive tape, sputter-coated (JEOL FFC-1100, Tokyo, Japan), and observed with SEM (JSM 840A, JEOL, Tokyo, Japan) at 15 kV acceleration voltage in secondary electrons mode at a magnification of 500×–50,000×.

### 2.6. Statistical Analysis

Statistical software (JMP 17.0, SAS Institute, Cary, NC, USA) was used to conduct the analyses. Means ± 1 standard deviation were calculated for DLS measurements. Means ± 1 standard error were calculated for the raw OD data from the MTT viability assay dataset. Using the Shapiro–Wilk test (*p* < 0.05), we confirmed that the data followed a normal distribution. Levene’s test (*p* < 0.05) confirmed that the variances were homogeneous. One-way analysis of variance (ANOVA) was performed considering an α value of 0.05. Tukey’s HSD was employed as a post hoc test (*p* < 0.05).

## 3. Results

The success of the CNC extraction was visually confirmed by the typical formation of birefringent domains when observed between crossed nicols, as reported by [28]. Then, the nano-size and the colloidal stability in water were assessed by dynamic light scattering (DLS) measurements, and we found a dimension of 92.4 ± 1.8 nm and a ζ-potential of −39.0 ± 1.0 mV (Figure 3a). Among the different CNC characteristics, particular attention was devoted to the surface. This is characterized by different hydroxyl groups of cellulose but also by negatively charged sulfate half-ester groups introduced by the sulfuric acid during the extraction procedure [29]. The presence of those groups on the CNC surface conferred the negative ζ-potential because sulfate half-esters are characterized by an extremely low pK_a_ and are completely hydrolyzed as anions in neutral water (Figure 3b). The half-sulfate ester groups seem to play an essential role in determining CNC properties and when CNCs are processed for different applications [30,31,32,33,34,35].

The FT-IR was used to determine the functional groups present in CNCs. The FT-IR spectrum is reported in Figure 3c: the broad band between 3400 and 3000 cm^−1^ is related to the intramolecular hydrogen-bonded stretching vibration of O-H (mainly the -OH groups on carbon 2, 3, and 6). On CNCs, unlike lignocellulosic raw materials, this band becomes narrower, longer, and more structured due to the high order of cellulose chain assembly. In fact, CNCs are also characterized by a high crystallinity index (>80%) where the diffraction pattern of cellulose I can be easily recognized [31]. The band at 2900 cm^−1^ indicates the C-H stretching vibration, while the small band at 1621 cm^−1^ indicates the presence of the C=O stretching vibration as well as the O-H bending vibration of absorbed water molecules. The spectrum is then dominated by the band at 1000 cm^−1^, which is associated with the C-O-C and C-OH stretching vibrations of the pyranose ring and secondary and primary alcohols [35].

SEM and TEM images are also reported in Figure 3d,e, respectively. The analysis confirmed the rod-like morphology of the nanoparticles and their dimensions, which was found to be roughly 200 nm. It is worth noting that the DLS method estimates the hydrodynamic diameter of particles approximated to spheres undergoing Brownian motion. Therefore, the nanoparticle dimensions made with DLS are observed to be about half as long as those made with SEM and TEM.

After the extensive CNC chemical–physical characterization, the colloidal suspensions were investigated in terms of biofilm formation inhibition, exploiting their peculiar surface chemistry. Initially, *S. mutans’* and *C. albicans’* adherence in response to the CNC solution tested at different concentrations showed a similar trend. The viable mass results indicated a significant inhibition of adherence on both *S. mutans* and *C. albicans*, starting from the lowest concentration tested (0.03%, *p* < 0.0001 and *p* = 0.008, respectively). This anti-adherent effect was much more pronounced on *S. mutans* (−70%) than on *C. albicans* (−30%). The effect reached maximum values at 0.25% and 0.5% concentrations for *S. mutans* and *C. albicans*, respectively. Complete inhibition was obtained for *S. mutans*, while a maximum of 80% inhibition was reached for *C. albicans* (Figure 4).

*S. mutans* biofilm formation (Figure 5a) was significantly reduced by CNCs when in a concentration of at least 0.5% (*p* = 0.01), reaching about −15%. Increasing the concentration did not produce additional reductions. Only the highest concentration of CNCs tested (2%, *p* = 0.01) reduced the artificial oral microcosm biofilm by about 25% (Figure 5b).

SEM observations are reported for *S. mutans* adherence (Figure 6) and biofilm formation (Figure 7) in the presence of CNC colloidal suspensions. A considerable reduction in adherence was found already, starting from the lowest concentration tested, which agrees with the viable biomass assay findings. Nevertheless, the very low number of microbial cells adhering to the surface did not show signs of distress (e.g., they were seen actively replicating and had intact membranes). Microbial nests were found all over the surface after 24 h (biofilm formation), showing active extracellular matrix production. A relatively lower presence of microbial cells was found when comparing the specimens treated with the maximum CNC concentration (2 wt. %) with the negative control. Again, microbial cells adhering to the surface did not show signs of distress. *C. albicans* adherence showed a comparable behavior to *S. mutans*, while no morphological differences in the artificial oral microcosm biofilm could be observed.

## 4. Discussion

There is currently an overwhelming interest in medical science regarding nanoparticles showing antimicrobial properties [7,30]. However, many of the possibilities entail concerns about their biocompatibility and toxicity [12,13,14]. Consequently, cellulose nanocrystals have received much attention due to their biocompatible and non-toxic properties, in addition to their promising antimicrobial properties. [14,15,31,32,33]. It is interesting to highlight the bio-based nature of this nano-technological platform and the possibility of extracting CNCs from almost all the lignocellulosic biomasses available, even agro-industrial by-products, making them particularly promising from a circular economy point of view [36,37].

This work used the selective and controlled hydrolysis of filter paper in concentrated H_2_SO_4_ to produce cellulose nanocrystals. Habibi et al. demonstrated in their literature review that the protocol used allowed for the selective acid hydrolysis of the amorphous cellulose areas, leading to the creation of rod-like nanocrystallites in a stable aqueous suspension [38]. For the first time, we evaluated in vitro the anti-adherent and anti-biofilm effect of CNCs against microbiological models of oral interest, including *S. mutans*, *C. albicans*, and the oral artificial microcosm made by mixed flora. Our results showed that the tested CNCs were already highly effective as an anti-adherent principle from the lowest tested concentration (0.03 wt. %). This activity was twice as high against *S. mutans* compared to *C. albicans*. SEM micrographs showed a very low number of cells adhering to the surface in the presence of CNCs, yet such cells did not show signs of tampering and were actively dividing. These findings suggest that the CNC mechanism of action is anti-adherent rather than one which directly inactivates microbial cells (Figure 8).

In fact, CNCs possess unique surface and colloidal properties, which makes them efficient flocculant agents in the separation of colloidal sized bacteria from their aqueous dispersion [39]. Efforts were made to utilize this property to develop disinfecting technology suitable for biomedical applications, particularly for biofilm prevention. Such efforts were directed toward exploiting interactions between CNCs and microbial cells that are based on depletion rather than an evident biocidal activity [39,40]. In agreement with our results, CNCs prevented microbial cells adhering to the substratum.

This finding is of utmost interest for an active principle in the oral environment. Adherence is an essential step for microbial colonization in the oral cavity [41,42]. Dental hard tissues are covered by an acquired salivary pellicle, which, from a microbial colonization point of view, makes them unique substrates in the human body. For this reason, the biofilm that permanently colonizes such an ecological niche possesses peculiar features [42]. Pioneer microorganisms that first adhere are equipped with adhesins that specifically bind to the acquired salivary pellicle, thus, resisting shear stresses [43]. Colonization by other taxa allows biofilm formation and maturation, including extracellular matrix production and an increase in its complexity to harbor at least 2000 taxa, with ~700 species residing in an individual’s mouth over a lifetime [22,44,45]. Some species are typically commensal and spend their time in symbiosis with the human host. Then again, some can cause oral diseases such as dental caries, gingivitis, and periodontitis [2,4,19,46]. Thus, control of microbial adherence in the oral cavity is essential for the control of biofilm and, ultimately, for the development of the above diseases [47]. For these reasons, our results offer a very promising possibility to control and modulate the interactions of oral microorganisms with the host and its substrata. Furthermore, the fact that the anti-adherent activity was higher for a pathogenic microorganism such as *S. mutans* than the other tested strain suggests the promising possibility of a selective activity that could be directed mainly towards pathogenic species. Such a hypothesis will be the core of future studies on the activity of CNCs in the oral environment.

The literature regarding the anti-adherent properties of CNCs is still scarce [30]. In 2017, it was demonstrated that 0.1 to 1 wt. % solutions of CNCs produced a very significant reduction in the adherence by *Escherichia coli* to the cell surfaces of an intestinal cell line. This effect was caused by a direct interaction of CNCs with microbial cells rather than a biocidal effect. In fact, a biocidal activity could be conferred to cellulose-based materials only by further functionalization with antimicrobial compounds [17,48]. Interestingly, Noronha et al. demonstrated significant biocidal activity of CNCs against *E. coli* due to a disruption of the integrity of lipid bilayer vesicles, assuming that CNCs penetrated the cells and inflicted irreversible damage to their membranes [49]. This result is in contrast with the findings of the present study. It must be noted that oral biofilms develop a thick layer of extracellular polysaccharides that protects microbial cells from the surrounding environment. For this reason, we never saw complete inactivation of microbial cells organized in biofilms when exposed to CNC concentrations up to 2 wt. %, and only a slight reduction in viability was observed, possibly due to the inactivation of the outer layer of cells. This consideration highlights the importance of the experimental conditions to replicate the physiological conditions as closely as possible. The latter include, for the oral environment, testing microbial cell colonization to natural hard surfaces in the presence of an acquired salivary pellicle and shear stresses that force adherent cells to present a biofilm structure and characteristics that resemble clinical ones [50,51]. In this sense, using a bioreactor allows the replication of most, if not all, of the clinically relevant biofilm behavior.

## 5. Conclusions

The results of the present study show that the tested CNCs exhibited excellent anti-adhesive activity against *S. mutans* and *C. albicans*. Such activity was significantly higher against *S. mutans* than *C. albicans*, suggesting a selective anti-adherent activity against oral pathogenic strains. At the same time, there was a minimal, albeit significant, anti-biofilm activity against both monospecies *S. mutans* biofilm and oral microcosm multispecies biofilm. This makes CNCs particularly interesting as anticaries agents, encouraging their use in the oral field. Most of the products currently used to control oral flora would benefit from using a bio-based nano-technological platform promptly available from almost all lignocellulosic biomasses. Such a platform could replace conventional biocidal compounds with easily obtainable CNCs that are promising from a circular economy point of view.

## Figures and Tables

**Figure 1 bioengineering-11-00355-f001:**
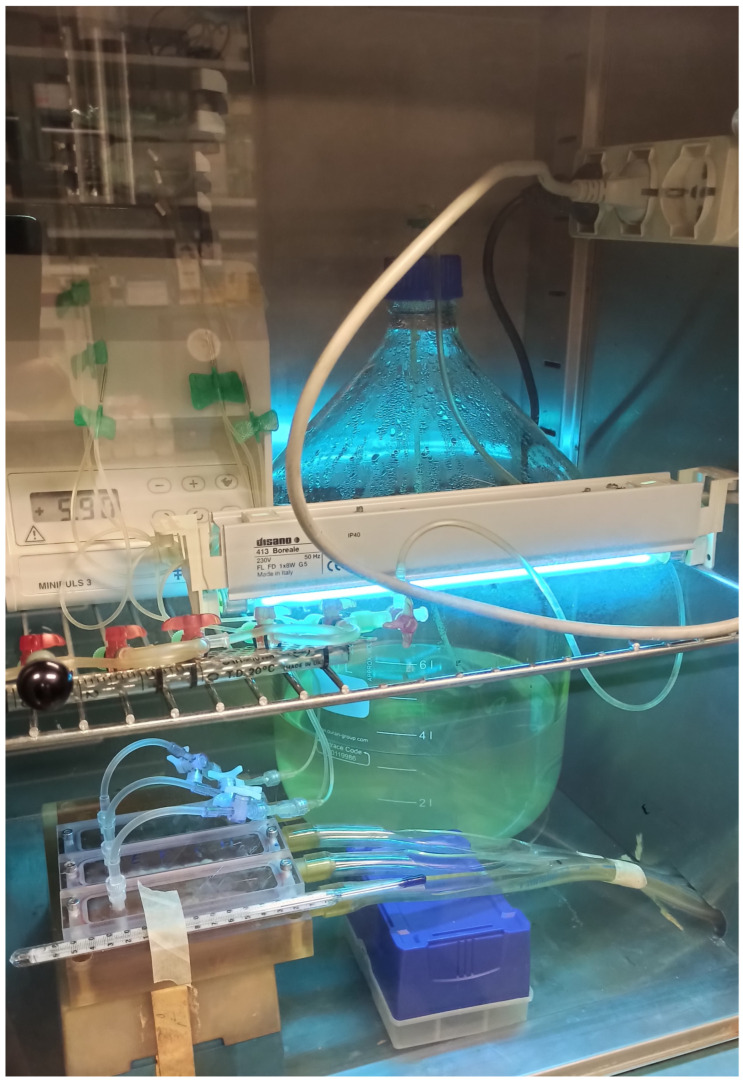
The M-DFR bioreactor functioning inside the incubator behind protective glass. An UV lamp is turned on to prevent back-contamination from the waste tubing to the bioreactor body. The peristaltic pump (upper left) provides a continuous flow of modified artificial saliva medium from the sterilized Pyrex bottle to the main bioreactor body, where it is distributed to the flow cells running in parallel.

**Figure 2 bioengineering-11-00355-f002:**
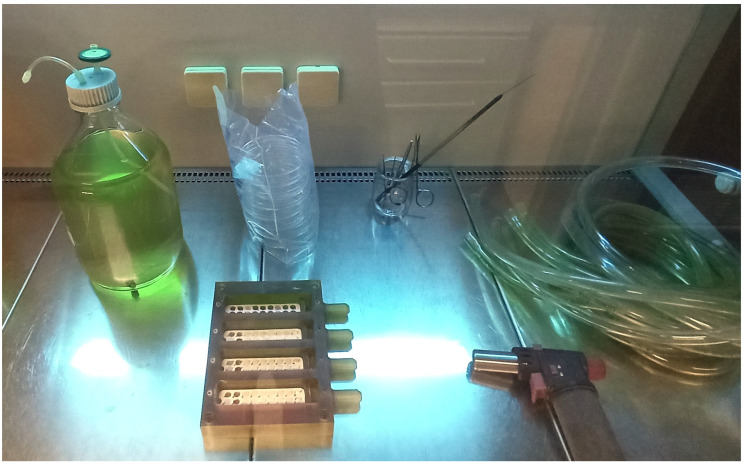
The M-DFR bioreactor prior to final assembly inside the sterile hood under UV irradiation. Its main body contains the enamel disks kept at the bottom of the parallel flow cells by the poly-tetrafluoroethylene customized specimen trays (white). The sterilized modified artificial saliva medium is seen on the left, while the tubing leading to the waste outside the incubator is shown on the right.

**Figure 3 bioengineering-11-00355-f003:**
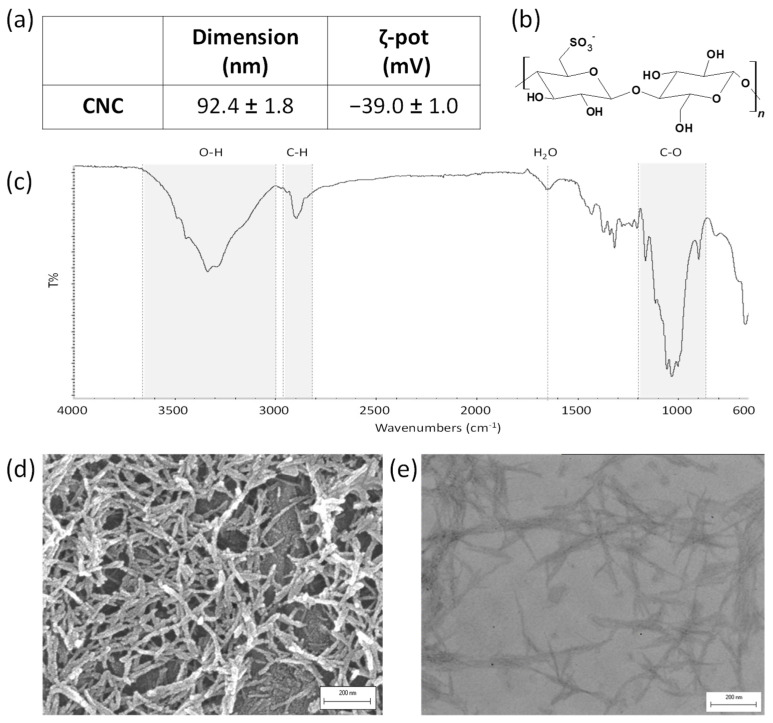
DLS measurements expressed as means ± 1 standard deviation (**a**), representative chemical structure of cellobiose unit (**b**), FT-IR spectrum (**c**), SEM image (**d**), and TEM image (**e**) of cellulose nanocrystals.

**Figure 4 bioengineering-11-00355-f004:**
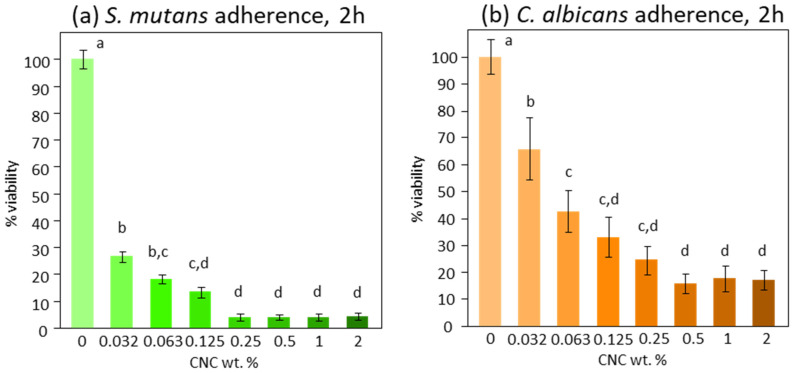
Viable microbial cells of *S. mutans* (**a**) and *C. albicans* (**b**) adherent to the pellicle-coated enamel surfaces after 2 h. Each bar shows a different CNC concentration in aqueous solution. Negative control (0% CNC) is set to 100% viability, while 0% viability was a 1 wt. % chlorhexidine digluconate solution. Different superscript letters indicate significant differences between groups (Tukey’s test, *p* < 0.05).

**Figure 5 bioengineering-11-00355-f005:**
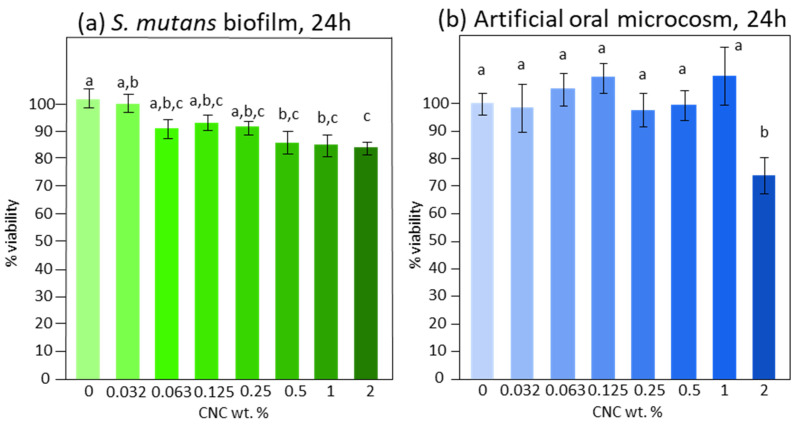
Viable biomass of *S. mutans* (**a**) and artificial oral microcosm from mixed plaque inoculum (**b**) developed under shear stress on the pellicle-coated enamel surfaces after 24 h. Each bar shows a different CNC concentration in aqueous solution. Negative control (0% CNC) is set to 100% viability, while 0% viability was a 1 wt. % chlorhexidine digluconate solution. Different superscript letters indicate significant differences between groups (Tukey’s test, *p* < 0.05).

**Figure 6 bioengineering-11-00355-f006:**
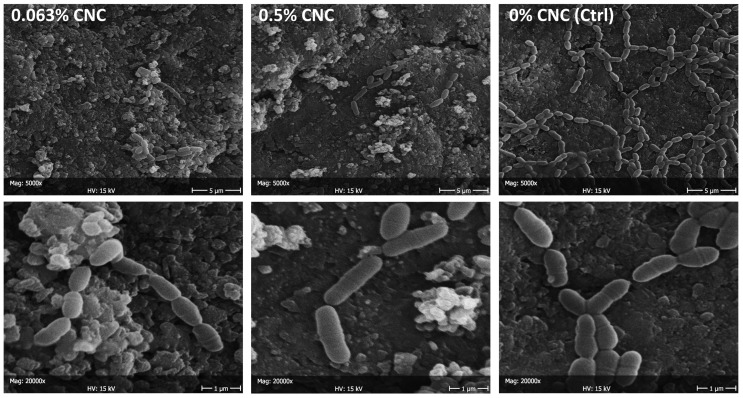
SEM micrographs at 5000× and 20,000× of *S. mutans* adhering to the specimens after 2 h (microbial adherence). The negative control (0% CNC) is compared with 0.063 wt. % (the minimum concentration, together with 0.032 wt. %, influencing adherence) and the concentration achieving the maximum effect, i.e., 0.5 wt. %. Whitish precipitates can be attributed to CNC clusters. The considerable reduction in adherence to an almost complete inhibition is consistent with the viable biomass assay findings. This suggests an interaction between CNC and microbial cells, preventing the latter from successfully adhering to the substrate.

**Figure 7 bioengineering-11-00355-f007:**
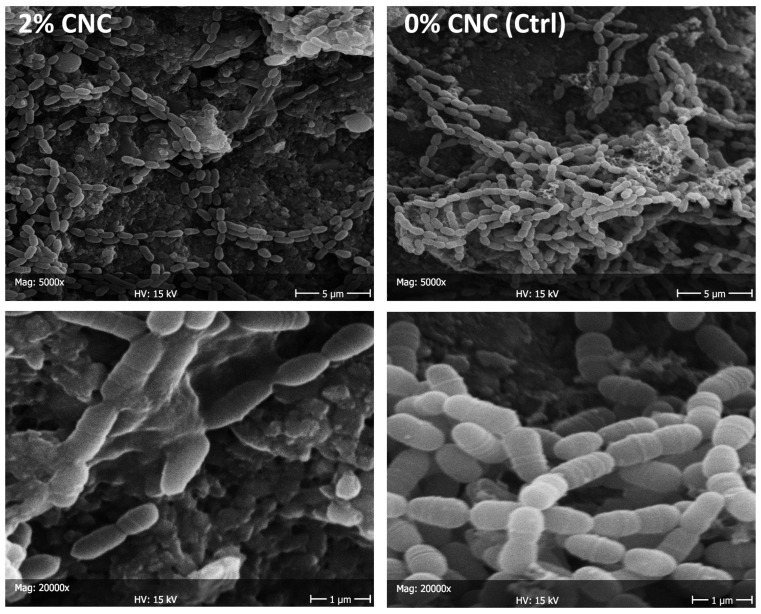
SEM micrographs at 5000× and 20,000× of *S. mutans* adhering to the specimens after 24 h (biofilm formation). The negative control (0% CNC) is compared with 2 wt. % (the concentration achieving the maximum effect). Whitish precipitates can be attributed to CNC clusters. Extracellular matrix can be seen as produced by microbial cells. Fewer cells are observed on the substrate after CNC treatment rather than the presence of cells showing signs of distress (e.g., low number of cells undergoing replication or non-intact membranes).

**Figure 8 bioengineering-11-00355-f008:**
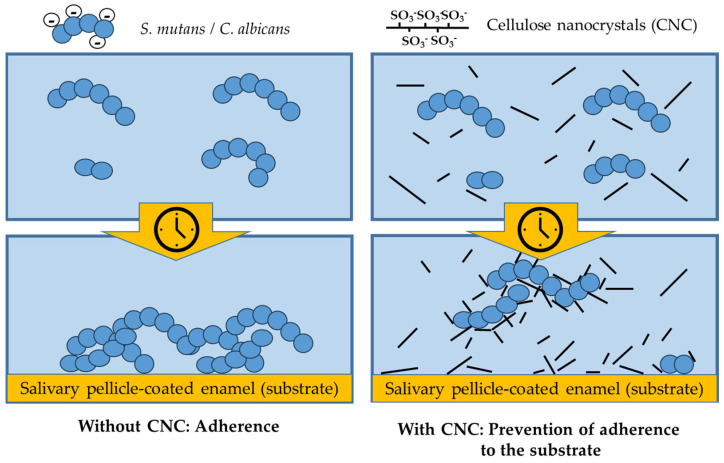
Diagram representing the interactions between CNCs and the tested oral microorganisms. Cellulose nanocrystals co-aggregate with microbial cells over time and prevent their interaction with the substrate, preventing microbial adherence without exerting any evident biocidal activity.

## Data Availability

All data are made available by the Corresponding Author (ACI) upon reasonable request.

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
