# Peer review of "Cellulose Nanocrystals Show Anti-Adherent and Anti-Biofilm Properties against Oral Microorganisms"

_bioengineering, 2024, doi:10.3390/bioengineering11040355_

Round 1

Reviewer 1 Report (Previous Reviewer 2)

Comments and Suggestions for Authors

The authors addressed most of my concerns carefully. I only have one minor comment.

The new Figure 8 is currently very vague in explaining the mechanism explaining. Currently, it is just drawing more black lines (as CNCs) on the right pannel. The figure needs to be self-explainable.

Author Response

Thank you for your comments. We modified Figure 8 so that the mechanism is more clear. Please see the attachment.

Reviewer 2 Report (Previous Reviewer 3)

Comments and Suggestions for Authors

The manuscript presents the results obtained from the studies regarding the ability of a cellulose nanocrystals (CNC) solution to reduce bacterial adhesion and biofilm formation using in vitro models. 

The authors try to reproduce in their study the different aspects of the oral environment.  Results obtained in this study show anti-adherent properties of the CNCs against S. mutans, suggesting a selective anti-adherent activity against pathogenic strains. The manuscript is well written, and experiments are conducted logically, with more details regarding the devices used. In my opinion,   this is the first study which tries to reproduce the environment which causes teeth damage and offers a potential solution to counteract this phenomenon.  For this reason, I recommend editors to publish this article.

As a minor observation, in my opinion, the sentence from rows 72-74 must be deleted or rewritten. This is because this sentence sounds like a... mathematical hypothesis. 

Author Response

Thank you very much, as per your suggestion, the sentence expressing the null hypothesis was removed.

Reviewer 3 Report (Previous Reviewer 4)

Comments and Suggestions for Authors

Thank you very much for sending me the manuscript again, I confirm that it is possible to publish.

Author Response

Thank you very much!

This manuscript is a resubmission of an earlier submission. The following is a list of the peer review reports and author responses from that submission.

Round 1

Reviewer 1 Report

Comments and Suggestions for Authors

iThenticate: The author, shows a similarity of 47%. Your manuscript should be revised to be below 20%.

It would be preferable if the author could provide a representation of the schematic diagram of your CNC against oral microorganisms as figure 1

Please re-check with the statistical analysis

The author should discuss the FTIR concentration that was used for the FIR study in figure 1?

In figure 2, display the percentage of viable cells in the vertical position along the y-axis.

I would appreciate it if you could proofread the paper for any grammatical errors.

In conclusion, you should write more on the relevance of your work.

Reviewer 2 Report

Comments and Suggestions for Authors

In this work, the authors conducted a preliminary evaluation of how CNC affects microbial activity and biofilm formation. However, the experiments are not enough to provide a clear logic chain that support the authors’ hypothesis. More evidence and investigations are needed to strengthen the results. I have the following questions.

1.     The DLS measurement of CNC is not appropriate. From the TEM image, it can clearly be seen that the length of CNC nanoparticle is above 200 nm. Also, the TEM image should be improved for particle size analysis.

2.     There are two Figure 1.

3.      The authors mentioned that “The quantity of adhering, viable, and  metabolically active biomass was represented by the relative absorbance in optical density (OD) units.” The colloidal CNC particle will affect the absorbance data. Can authors explain the reliability of their results?

4.     The authors need to clearly explain their methodology on the evaluation of bacteria adherence.

5.     What is the proposed mechanism of CNC being anti-bacteria? And how would the authors prove it?

6.     Why is there no significant change in the viable biomass with the presence of CNC, which is contradictory to the adherence tests?

7.     The amount of evidence is insufficient to support the results (Only two main figures). The authors need to provide more obvious and consistent results to demonstrate their hypothesis.

Reviewer 3 Report

Comments and Suggestions for Authors

In their studies, the authors evaluate the ability of a solution of cellulose nanocrystals (CNS) to reduce bacterial adhesion and biofilm formation, using in vitro tests. For this, the authors used two microbiological models that reproduce the human oral environment. The results obtained showed that the CNCs tested showed an antiadhesive activity against S. mutans and C. albicans, significantly higher against S. mutans than C. albicans. This fact indicates a selective anti-adherence activity against certain strains of oral pathogenic microorganisms. At the same time, an anti-biofilm activity was highlighted, for S. mutans and respectively for the biofilm determined by the multispecies in the oral microcosm. 

The manuscript is well written, experiments are conducted correctly and logically and the results and discussion are well written. The results obtained in this study recommend CNCs as promising antiaging agents, and this is the novelty of this experimental study and for this reason, I recommend its publication.  However, some minor changes are required before publication as follows:

1) For devices appearing in lines R73, R81, R87 and R149 authors must provide the city and country where they were produced. Additionally, in the Appendix, authors must provide an image of the modified device, which appears in the manuscript on line 149.

2) Wavelength units must be superscripted (line 74)

3) A sentence cannot start with an abbreviation. So the sentence starting at line 83 needs to be rewritten.

4) On line 118, the title of chapter 2.3.2 must be modified; instead of "Bacteria" it should be written "Microorganisms".

5) On page 7, line 251, instead of Figure 1, must be written: Figure 2a,b

6) On the same page in line 253, instead of "Figure 1" must written: Figure 2;

7) On line 258, instead of "Figure 2a" must  be written: Figure 3a;

8) On line 261, instead of "Figure 2b" must be written: Figure 3b;

9) On line 263, instead of "Figure 2" must be written: Figure 3.

Reviewer 4 Report

Comments and Suggestions for Authors

Manuscript of considerable interest for the dental sector, requires a major revision before being able to proceed with the evaluation of a possible publication.

Abstract: highlight the results obtained by inserting statistical analyzes and not percentages.

Keywords; very generic, add specific ones that are registered on MeSH.

Introduction: Add all predisposing factors that induce the incidence of caries. (Butera et al.)

Materials and methods well described

Very confusing results, the images are very beautiful in high resolution, please reorganize the tables so that they are usable for the reader.

Discussion: set as future objectives, given that they are equally animal waste, the use of postbiotics to evaluate the reduction of biofilm and whether it acts equally on S. Mutans and C. albicans. (Scribante et al)

Reformulate the conclusions based on the requested changes.

Add references to the bibliography